# CRF-CNN: Modeling Structured Information in Human Pose Estimation

**Xiao Chu**
The Chinese University of Hong Kong
xchu@ee.cuhk.edu.hk

**Wanli Ouyang**
The Chinese University of Hong Kong
wlouyang@ee.cuhk.edu.hk

**Hongsheng Li**
The Chinese University of Hong Kong
hsli@ee.cuhk.edu.hk

**Xiaogang Wang**
The Chinese University of Hong Kong
xgwang@ee.cuhk.edu.hk

## Abstract

Deep convolutional neural networks (CNN) have achieved great success. On the other hand, modeling structural information has been proved critical in many vision problems. It is of great interest to integrate them effectively. In a classical neural network, there is no message passing between neurons in the same layer. In this paper, we propose a CRF-CNN framework which can simultaneously model structural information in both output and hidden feature layers in a probabilistic way, and it is applied to human pose estimation. A message passing scheme is proposed, so that in various layers each body joint receives messages from all the others in an efficient way. Such message passing can be implemented with convolution between features maps in the same layer, and it is also integrated with feedforward propagation in neural networks. Finally, a neural network implementation of end-to-end learning CRF-CNN is provided. Its effectiveness is demonstrated through experiments on two benchmark datasets.

## 1 Introduction

A lot of efforts have been devoted to structure design of convolutional neural network (CNN). They can be divided into two groups. One is to achieve higher expressive power by making CNN deeper [19, 10, 20]. The other is to model structures among features and outputs, either as post processing [6, 2] or as extra information to guide the learning of CNN [29, 22, 24]. They are complementary.

Human pose estimation is to estimate body joint locations from 2D images, which could be applied to assist other tasks such as [4, 14, 26] The very first attempt adopting CNN for human pose estimation is DeepPose [23]. It used CNN to regress joint locations repeatedly without directly modeling the output structure. However, the prediction of body joint locations relies both on their own appearance scores and the prediction of other joints. Hence, the output space for human pose estimation is structured. Later, Chen and Yuille [2] used a graphical model for the spatial relationship between body joints and used it as post processing after CNN. Learning CNN features and structured output together was proposed in [22, 21, 24]. Researchers were also aware of the importance of introducing structures at the feature level [3]. However, the design of CNN for structured output and structured features was heuristic, without principled guidance on how information should be passed. As deep models are shown effective for many practical applications, researchers on statistical learning and deep learning try to use probabilistic models to illustrate the ideas behind deep models [9, 7, 29].

Motivated by these works, we provide a CRF framework that models structures in both output and hidden feature layers in CNN, called CRF-CNN. It provides us with a principled illustration on how to model structured information at various levels in a probabilistic way and what are the assumptions

made when incorporating different CRF into CNN. Existing works can be illustrated as special implementations of CRF-CNN. DeepPose [23] only considered the feature-output relationship, and the approaches in [2, 22] considered feature-output and output-output relationships. In contrast, our proposed full CRF-CNN model takes feature-output, output-output, and feature-feature relationships into consideration, which is novel in pose estimation.

It also facilitates us in borrowing the idea behind the sum-product algorithm and developing a message passing scheme so that each body joint receives messages from all the others in an efficient way by saving intermediate messages. Given a set of body joints as vertices on a graph, there is no conclusion on whether a tree structured model [28, 8] or a loopy structured model [25, 16] is the best choice. A tree structure has exact inference while a loopy structure can model more complex relationship among vertices. Our proposed message passing scheme is applicable to both.

Our contributions can be summarized as follows. (1) A CRF is proposed to simultaneously model structured features and structured body part spatial relationship. We show step by step how approximations are made to use an end-to-end learning CNN for implementing such CRF model. (2) Motivated by the efficient algorithm for marginalization on tree structures, we provide a message passing scheme for our CRF-CNN so that every vertex receives messages from all the others in an efficient way. Message passing can be implemented with convolution between feature maps in the same layer. Because of the approximation used, this message passing can be used for both tree and loopy structures. (3) CRF-CNN is applied to two human pose estimation benchmark datasets and achieve better performance on both dataset compared with previous methods.

## 2  CRF-CNN

The power of combing powerful statistical models with CNN has been proved [6, 3]. In this section we start with a brief review of CRF and study how the pose estimation problem can be formulated under the proposed CRF-CNN framework. It includes estimating body joints independently from CNN features, modeling the spatial relationship of body joints in the output layer of CNN, and modeling the spatial relationship of features in the hidden layers of CNN.

Let $\mathbf{I}$ denote an image, and $\mathbf{z} = \{\mathbf{z}_1, ..., \mathbf{z}_N\}$ denote locations of $N$ body joints. We are interested in modeling the conditional probability $p(\mathbf{z}|\mathbf{I}, \Theta)$ parameterized by $\Theta$, expressed in a Gibbs distribution:

$$p(\mathbf{z}|\mathbf{I}, \Theta) = \frac{e^{-En(\mathbf{z}, \mathbf{I}, \Theta)}}{Z} = \frac{e^{-En(\mathbf{z}, \mathbf{I}, \Theta)}}{\sum_{\mathbf{z} \in \mathcal{Z}} e^{-En(\mathbf{z}, \mathbf{I}, \Theta)}}, \tag{1}$$

where $En(\mathbf{Z}, \mathbf{I}, \Theta)$ is the energy function. The conditional distribution by introducing latent variables $\mathbf{h} = \{h_1, h_2, \dots, h_K\}$ can be modeled as follows:

$$p(\mathbf{z}|\mathbf{I}, \Theta) = \sum_{\mathbf{h}} p(\mathbf{z}, \mathbf{h}|\mathbf{I}, \Theta), \text{where } p(\mathbf{z}, \mathbf{h}|\mathbf{I}, \Theta) = \frac{e^{-En(\mathbf{z}, \mathbf{h}, \mathbf{I}, \Theta)}}{\sum_{\mathbf{z} \in \mathcal{Z}, \mathbf{h} \in \mathcal{H}} e^{-En(\mathbf{z}, \mathbf{h}, \mathbf{I}, \Theta)}} \tag{2}$$

$En(\mathbf{z}, \mathbf{h}, \mathbf{I}, \Theta)$ is the energy function to be defined later. The latent variables correspond to features obtained from a neural network in our implementation. We define an undirected graph $\mathcal{G} = (\mathcal{V}, \mathcal{E})$, where $\mathcal{V} = \mathbf{z} \cup \mathbf{h}$, $\mathcal{E} = \mathcal{E}_z \cup \mathcal{E}_h \cup \mathcal{E}_{zh}$. $\mathcal{E}_z$, $\mathcal{E}_h$, and $\mathcal{E}_{zh}$ denote sets of edges connecting body joints, connecting latent variables, and connecting latent variables with body joints, respectively.

### 2.1  Model 1

Denote $\emptyset$ as an empty set. If we suppose there is no edge connecting joints and no edge connecting latent variables in the graphical model, i.e. $\mathcal{E}_z = \emptyset$, $\mathcal{E}_h = \emptyset$, then

$$p(\mathbf{z}, \mathbf{h}|\mathbf{I}, \Theta) = \prod_i p(\mathbf{z}_i|\mathbf{h}, \mathbf{I}, \Theta) \prod_k p(h_k|\mathbf{I}, \Theta), \tag{3}$$

$$En(\mathbf{z}, \mathbf{h}, \mathbf{I}, \Theta) = \sum_{(i,k) \in \mathcal{E}_{zh}} \psi_{zh}(\mathbf{z}_i, h_k) + \sum_k \phi_h(h_k, \mathbf{I}), \tag{4}$$

where $\phi_h(*)$ denotes the unary/data term for image $\mathbf{I}$, $\psi_{zh}(*, *)$ denotes the terms for the correlations between latent variables $\mathbf{h}$ and body joint configurations $\mathbf{z}$. It corresponds to the model in Fig. 1(a) and it is a typical feedforward neural network.

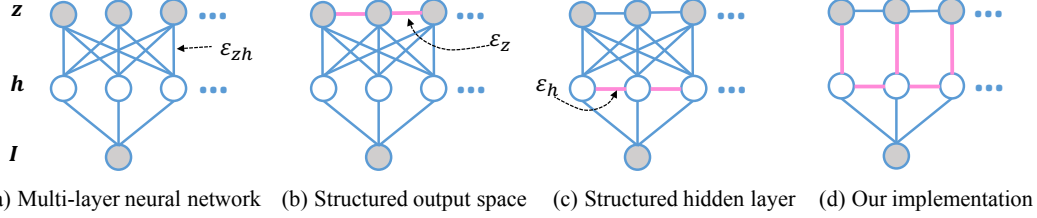

| (a) Multi-layer neural network | (b) Structured output space | (c) Structured hidden layer | (d) Our implementation |

Figure 1: Different implementations of the CRF-CNN framework.

*Example.* In DeepPose [23], CNN features $\mathbf{h}$ in the top hidden layer were obtained from images, and could be treated as latent variables and illustrated by term $\phi_h(h_k, \mathbf{I})$ in (4). There is no connection between neurons in hidden layers. Body joint locations were estimated from CNN features in [23], which could be illustrated by the term $\psi_{zh}(\mathbf{z}_i, h_k)$. The body joints are independently estimated without considering their correlations, which means $\mathcal{E}_z = \emptyset$.

## 2.2  Model 2

If we suppose $\mathcal{E}_h = \emptyset$ in the graphical model, $p(\mathbf{z}, \mathbf{h}|\mathbf{I}, \Theta)$ becomes

$$p(\mathbf{z}, \mathbf{h}|\mathbf{I}, \Theta) = p(\mathbf{z}|\mathbf{h}, \mathbf{I}, \Theta) \prod_k p(h_k|\mathbf{I}, \Theta). \tag{5}$$

Compared with (3), joint locations are no longer independent. The energy function for this model is

$$En(\mathbf{z}, \mathbf{h}, \mathbf{I}, \Theta) = \sum_{\substack{(i,j)\in\mathcal{E}_z \\ i<j}} \psi_z(\mathbf{z}_i, \mathbf{z}_j) + \sum_{(i,k)\in\mathcal{E}_{zh}} \psi_{zh}(\mathbf{z}_i, h_k) + \sum_k \phi_h(h_k, \mathbf{I}). \tag{6}$$

It corresponds to the model in Fig. 1(b). Compared with (4), $\psi_z(\mathbf{z}_i, \mathbf{z}_j)$ in (6) is added to model the pairwise relationship between joints.

*Example.* To model the spatial relationship among body joints, the approaches in Yang *et al.* [22] built up pairwise terms and spatial models. They are different implementations of $\psi_z(\mathbf{z}_i, \mathbf{z}_j)$ in (6).

## 2.3  Our model

In our model, $\mathbf{h}$ is considered as a set of discrete latent variables and each $h_k$ is represented as a 1-of-$L$ $L$ dimensional vector. $p(\mathbf{z}, \mathbf{h}|\Theta)$ and $En(\mathbf{z}, \mathbf{h}, \mathbf{I}, \Theta)$ for this model are:

$$p(\mathbf{z}, \mathbf{h}|\mathbf{I}, \Theta) = p(\mathbf{z}|\mathbf{h}, \mathbf{I}, \Theta)p(\mathbf{h}|\mathbf{I}, \Theta). \tag{7}$$

$$En(\mathbf{z}, \mathbf{h}, \mathbf{I}, \Theta) = \sum_{\substack{(k,l)\in\mathcal{E}_h \\ k<l}} \psi_h(h_k, h_l) + \sum_{\substack{(i,j)\in\mathcal{E}_z \\ i<j}} \psi_z(\mathbf{z}_i, \mathbf{z}_j) + \sum_{(i,k)\in\mathcal{E}_{zh}} \psi_{zh}(\mathbf{z}_i, h_k) + \sum_k \phi_h(h_k, \mathbf{I}). \tag{8}$$

It is the model in Fig. 1(c) and exhibits the largest expressive power compared with the models in (4) and (6). $\psi_h(h_k, h_l)$ is added to model the pairwise relationship among features/latent variables in (8).

*Details on the set of edges $\mathcal{E}$.* Body joints have structures and it may not be suitable to use a fully connected graph. The tree structure in Fig. 2(b) is widely used since it fits human knowledge on the skeleton of body joints and how body parts articulate. A further benefit for a tree structure with $N$ vertices is that all vertices can receive messages from others with $2N$ message passing operations. To better define the structure of latent variables $\mathbf{h}$, we group the latent variables so that a joint $\mathbf{z}_i$ corresponds to a particular group of latent variables denoted by $\mathbf{h}_i$, and $\mathbf{h} = \cup_i \mathbf{h}_i$. $\sum_{(i,k)\in\mathcal{E}_{zh}} \psi_{zh}(\mathbf{z}_i, h_k)$ in (8) is simplified into $\sum_{i=1}^N \psi_{zh}(\mathbf{z}_i, \mathbf{h}_i)$, i.e. $\mathbf{z}_i$ is only connected to latent variables in $\mathbf{h}_i$. We further constrain connections among feature groups: $(\mathbf{h}_i, \mathbf{h}_j) \in \mathcal{E}_h \iff$

$(\mathbf{z}_i, \mathbf{z}_j) \in \mathcal{E}_z$. It means that feature groups are connected if and only if their corresponding body joints are connected. Fig. 1(d) shows an example of this model. Our implementation is as follows:

$$En(\mathbf{z}, \mathbf{h}, \mathbf{I}, \Theta) = \sum_{\substack{(i,j) \in \mathcal{E}_h \\ i < j}} \psi_h(\mathbf{h}_i, \mathbf{h}_j) + \sum_{\substack{(i,j) \in \mathcal{E}_z \\ i < j}} \psi_z(\mathbf{z}_i, \mathbf{z}_j) + \sum_{i=1}^{N} \psi_{zh}(\mathbf{z}_i, \mathbf{h}_i) + \sum_{k=1}^{K} \phi_h(h_k, \mathbf{I}), \quad (9)$$

## 3  Implementation with neural networks

In order to marginalize latent variables $\mathbf{h}$ and obtain $p(\mathbf{z}|\mathbf{I}, \Theta)$, the computational complexity of marginalization in (2) is high, exponentially proportional to the cardinality of $\mathbf{h}$. In order to infer $p(\mathbf{z}|\mathbf{I}, \Theta)$ in a more efficient way, we use the following approximations:

$$p(\mathbf{z}|\mathbf{I}, \Theta) = \sum_{\mathbf{h}} p(\mathbf{z}, \mathbf{h}|\mathbf{I}, \Theta) = \sum_{\mathbf{h}} p(\mathbf{z}|\mathbf{h}, \mathbf{I}, \Theta) p(\mathbf{h}|\mathbf{I}, \Theta) \approx p(\mathbf{z}|\tilde{\mathbf{h}}, \mathbf{I}, \Theta), \quad (10)$$

$$\text{where } \tilde{\mathbf{h}} = [\tilde{\mathbf{h}}_1, \tilde{\mathbf{h}}_2, \dots, \tilde{\mathbf{h}}_N] = E[\mathbf{h}] = \sum_{h} \mathbf{h} p(\mathbf{h}|\mathbf{I}, \Theta), \quad (11)$$

In (10) and (11), we replace $\mathbf{h}$ by its average configuration $\tilde{\mathbf{h}} = E[\mathbf{h}]$ and this approximation was also used in greedy layer-wise learning for deep belief net in [11].

$$p(\mathbf{h}|\mathbf{I}, \Theta) \approx \prod_i Q(\mathbf{h}_i|\mathbf{I}, \Theta), \quad (12)$$

$$Q(\mathbf{h}_i|\mathbf{I}, \Theta) = \frac{1}{Z_{h,i}} \exp \left\{ - \sum_{h_k \in \mathbf{h}_i} \phi_h(h_k, \mathbf{I}) - \sum_{\substack{(i,j) \in \mathcal{E}_h \\ i < j}} \psi_h(\mathbf{h}_i, Q(\mathbf{h}_j|\mathbf{I}, \Theta)) \right\}. \quad (13)$$

The target is to marginalize the distribution of $h$, as shown in 12. We adopt the classical mean-field approximation approach for message passing[15]. $p(\mathbf{h}|I, \Theta)$ in (11) is approximated by a product of independent $Q(\mathbf{h}_i|I, \Theta)$ in (12) and (13).

We first ignore the pairwise term $\psi_h(\mathbf{h}_i, \mathbf{h}_j)$ which will be addressed later in Section 3.1. Suppose $\phi_h(h_k, \mathbf{I}) = h_k \mathbf{w}_k^{\mathsf{T}} \mathbf{f}$, where $\mathbf{f}$ is the feature representation of image $\mathbf{I}$. For a binary latent variable $h_k$,

$$\tilde{h}_k = E[h_k] = \sum_{h_k} h_k Q(h_k|I, \Theta) = \text{sigm}(\phi_h(h_k, \mathbf{I})) = \text{sigm}(\mathbf{w}_k^{\mathsf{T}} \mathbf{f}), \quad (14)$$

where $\text{sigm}(x) = 1/(1 + e^{-x})$ is the sigmoid function. Therefore, the mapping from $\mathbf{f}$ to $\tilde{\mathbf{h}}$ can be implemented with one-layer transformation in a neural network and sigmoid is the activation function. $\tilde{\mathbf{h}}$ is a new feature vector derived from $\mathbf{f}$ and $\mathbf{f}$ can be obtained from lower layers in a network.

### 3.1  Message passing on tree structured latent variables

In order to infer $p(\mathbf{z}|\mathbf{I}, \Theta)$, the key challenge in our framework is to obtain the marginalized distribution of hidden units, *i.e.* , $Q(\mathbf{h}_i|\mathbf{I}, \Theta)$ in (12). One can obtain $Q(\mathbf{h}_i|\mathbf{I}, \Theta)$ through message passing and further estimate $\tilde{\mathbf{h}}$ Then $p(\mathbf{z}|\tilde{\mathbf{h}}, \mathbf{I}, \Theta)$ in (10) can be estimated with existing works such as [2, 28].

According to the sum-product algorithm for a tree structure, every node can receive the messages from other nodes through two message passing routes, first from leaves to a root and then from the root to the leaves [13]. The key is to have a planned route and to store the intermediate messages. Our proposed messaging passing algorithm is summarized in Algorithm 1. An example of message passing for a tree structure with 4 nodes as shown in Fig. 2(c). For detailed illustrations of 2, please refer to the supplementary material.

We drop $\mathbf{I}$ and $\Theta$ to be concise.

---

**Algorithm 1** Message passing among features on factor graph.

---

1: **procedure** BELIEF PROPAGATION($\Theta$)
2:     $U_k \leftarrow \mathbf{f} \otimes \mathbf{w}_k$, for $k = 1$ to $K$                                             $\triangleright$ Initialization
3:     **for** $m = 1$ to $M$ **do**                                                           $\triangleright$ Passing messages $M$ times
4:         Select a predefined message passing route $\mathbb{S}_m$
5:         **for** $e = 1$ to $|\mathcal{E}_h|$ **do**
6:             Choose an edge $(j \rightarrow k)$ from $\mathcal{E}_h$ according to the route $\mathbb{S}_m$
7:             Denote $ne(j)$ as the set of neighboring nodes for node $j$ on the graphical model
8:             **if** $k$ is a factor node denoted by $f_k$ **then**
9:                 $F_{j \rightarrow f_k} \leftarrow U_j + \sum_{f_p \in ne(j) \setminus k} F_{f_p \rightarrow j}$      $\triangleright$ Pass message from factors to variable
10:                 $Q_{j \rightarrow f_k} \leftarrow \tau(-F_{j \rightarrow k})$                                    $\triangleright$ Normalize
11:            **else**
12:                 Denote the factor node $j$ by $f_j$
13:                 $F_{f_j \rightarrow k} \leftarrow \sum_{p \in par(j) \setminus k} Q_{p \rightarrow f_j} \otimes \mathbf{w}^{p \rightarrow k}$      $\triangleright$ Pass message to the factor
14:            **end if**
15:        **end for**
16:    **end for**
17:    **for** $k = 1$ to $K$ **do**
18:        $Q(h_k) \leftarrow \tau(U_k + \sum_{f_p \in ne(k)} F_{f_p \rightarrow k})$
19:    **end for**
20: **end procedure**

---

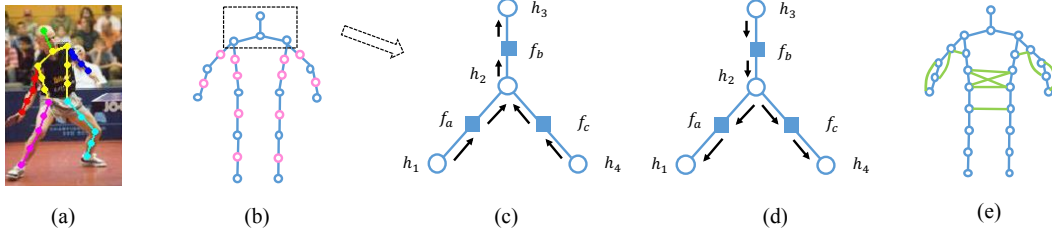

(a)             (b)             (c)             (d)             (e)

Figure 2: Message passing. (a) is the annotation of a person with its tree structure. (b) is the tree structured model employed on the LSP dataset. In (b), the pink colored nodes are linearly interpolated. (c,d) show message passing on a factored graph with different routes. (e) is a loopy model. In (e), the edges in green color are extra edges added on the tree structured model in(b).

According to the mean-field approximation algorithm, the above message passing process should be conducted for multiple times with share parameter to converge. To implement $\psi_h(h_i, h_j)$, we use matrix multiplication for easier illustration but convolution (which is a special form of matrix multiplication) for implementation in Algorithm 1. Then message passing is implemented with convolution between feature maps.

The proposed method is extensible to loopy structured graphs, as shown in Fig. 2(e). The underlying concept of building up probabilistic model at feature level is the same. However, for loopy structures, the key challenge is to define the rule in message passing. Either a sequence of asymmetric message passing order is predefined, which seems not reasonable for symmetric structure of human poses, or use the flooding scheme to repeated collect information for neighborhood joints. We compared tree structure with loop structure with flooding scheme in the experimental section.

## 3.2    Overall picture of CRF-CNN for human pose estimation

An overview of the approach is shown in Fig. 3. In this pipeline, the prediction on $i$th body part configuration $\mathbf{z}_i$ is represented by a score map $p(\mathbf{z}_i|\mathbf{h}) = \{\tilde{z}_i^{(1,1)}, \tilde{z}_i^{(1,2)}, \ldots\}$, where $\tilde{z}_i^{(x,y)} \in [0,1]$ denotes the predicted confidence on the existence of the $i$th body joint at the location $(x, y)$. Similarly, the group of features $\tilde{\mathbf{h}}_i$ used for estimating $p(\mathbf{z}|\mathbf{h})$ is represented by $\tilde{\mathbf{h}}_i = \{\tilde{h}_i^{(1,1)}, \tilde{h}_i^{(1,2)}, \ldots\}$,

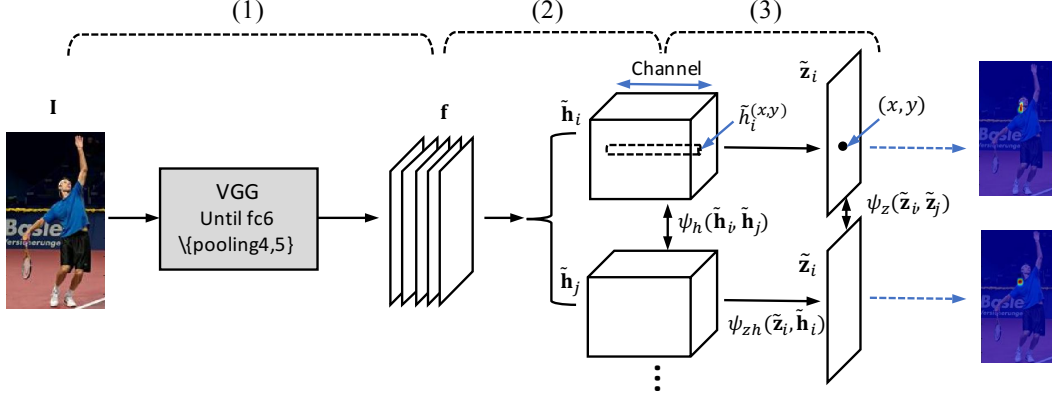

Figure 3: CNN implementation of our model. (1) We use the fc6 layer of VGG to obtain features $\mathbf{f}$ from an image. (2) The features $\mathbf{f}$ are then used for passing messages among latent variables $\mathbf{h}$. (3) Then the estimated latent variables $\tilde{\mathbf{h}}$ are used for estimating the predicted body part score maps $\tilde{\mathbf{z}}$. We only show the message passing process between two joints to be concise. Best viewed in color.

$i = 1, \ldots, N$. $\tilde{h}_i^{(x,y)}$ is a length-$L$ vector. Therefore, the feature group $\tilde{\mathbf{h}}_i$ is represented by a feature map of $L$ channels, where $h_i^{(x,y)}$ contains $L$ channels of features at location $(x, y)$.

1) It comprises a fully convolutional network stage, which takes an image as input and outputs features $\mathbf{f}$. We use the fully convolutional implementation of VGG and the output of fc6 in VGG is used as the feature map $\mathbf{f}$.

2) Messages are passed among features $\mathbf{h}$ with Algorithm 1. Initially, data term $U_k$ for the $k$th feature group is obtained from feature map $\mathbf{f}$ by convolution, which is our implementation of term $\phi_h(h_k, \mathbf{I})$ in (13) and corresponds to Algorithm 1 line 2. Then CNN is used for passing messages among $\mathbf{h}$ using lines 3-19 in Algorithm 1, which implements term $\psi_h(\mathbf{h}_i, Q(\mathbf{h}_j|\mathbf{I}, \Theta))$ in (13) by convolution. After message passing, the $\tilde{\mathbf{h}}_i$ for $i = 1, \ldots, N$ is obtained and treated as feature maps to be used.

3) Then the feature maps $\tilde{\mathbf{h}}_i$ for $i = 1, \ldots, N$ are used to obtain the score map for inferring $p(\mathbf{z}|\mathbf{h}, \mathbf{I}, \Theta)$ with (10). As a simple example for illustration, we can use $\tilde{z}_i^{(x,y)} = p\left(z_i^{(x,y)} = 1|\mathbf{h}_i, \mathbf{I}\right) = \text{sigm}\left(\mathbf{w}_i^{\text{T}} \tilde{h}_i^{(x,y)}\right)$ to obtain the predicted score $\tilde{z}_i^{(x,y)}$ for the $i$th part at location $(x, y)$. In this case, $\tilde{h}_i^{(x,y)}$ is the feature with $L$ channels at location $(x, y)$ and $\mathbf{w}_i$ can be treated as the classifier. Our implementation uses the approach in [2] to infer $p(\mathbf{z}|\mathbf{h}, \mathbf{I}, \Theta)$, which also models the spatial relationship among $\mathbf{z}_i$.

During training, a whole image (or many of them) can be used as the mini-batch and the error at each output location of the network can be computed using an appropriate loss function with respect to the ground truth of the body joints. We use softmax loss with respect to the estimated part configuration $\mathbf{z}$ as the approximate loss function. Since we have used CNN from input to features $\mathbf{f}$, $\tilde{\mathbf{h}}_i$ and $\tilde{\mathbf{z}}$, a single CNN is used for obtaining the score map of body joints from the image. End-to-end learning with softmax loss and standard BP is used.

## 4 Experiment

We conduct experiments on two benchmark datasets: the LSP dataset [12] and the FLIC dataset [18]. LSP contains $2,000$ images. $1,000$ images for training and $1,000$ for test. Each person is annotated with 14 joints. FLIC contains $3,987$ training images and $1,016$ testing images from Hollywood movies with upper body annotated. On both datasets, we use observer centric annotation for training and evaluation. We also use negative samples, *i.e.* images not containing any person, from the INRIA dataset [5]. In summary, we are consistent with Chen *et al.* [2] in training data preparation.

## 4.1 Results on the LSP dataset

The experimental results for our and previous approaches on LSP are shown in Table 1. For evaluation metric, we choose the prevailing evaluation method: *strict* Percentage of Correct Parts (PCP). Under this metric, a limb is considered to be detected only if both ends of an limb lie in 50% of the length *w.r.t.* the ground-truth location. For pose estimation, it is well known that the accuracy of CNN features is higher than handcrafted features. Therefore, we only compare with methods that use CNN features to be concise. Pishchulin *et al.* [17] use extra training data, so we do not compare with it. Yang *et al.* [27] learned features and structured body part configurations simultaneously. Our performance is better than them because we model structure among features. Chu *et al.* [3] learned structured features and heuristically defined a message passing scheme. Using only the LSP training data, these two approaches have the highest PCP (Observer-Centric) reported in [1]. The model in [3] has no probabilistic interpretation and cannot be modeled as CRF. Most vertices in their CNN can only receive information from half of the vertices, while in our message passing scheme each node could receive information from all vertices, since it is developed from CRF and the sum-product algorithm. The approaches in [27, 3] are all based on the VGG structure as ours. By using a more effective message passing scheme, our method reduces the mean error rate by 10%.

Table 1: Quantitative results on LSP dataset (PCP)

| Experiment | Torso | Head | U.arms | L.arms | U.legs | L.legs | **Mean** |
|---|---|---|---|---|---|---|---|
| Chen&Yuille [2] | 92.7 | 87.8 | 69.2 | 55.4 | 82.9 | 77.0 | 75.0 |
| Yang *et al.* [27] | **96.5** | 83.1 | 78.8 | 66.7 | 88.7 | 81.7 | 81.1 |
| Chu *et al.* [3] | 95.4 | 89.6 | 76.9 | 65.2 | 87.6 | 83.2 | 81.1 |
| Ours | 96.0 | **91.3** | **80.0** | **67.1** | **89.5** | **85.0** | **83.1** |

## 4.2 Results on the FLIC dataset

We use Percentage of Correct Keypoints (PCK) as the evaluation metric. Because it is widely adopted by previous works on FLIC, it provides convenience for comparison. These published works only reported results on elbow and wrist and we follow the same practice. PCK reports the percentage of predictions that lay in the normalized distance of annotation. Toshev *et al.* [23], Chen and Yuille [2] and Tompson *et al.* [21] also used CNN features. When compared with previous state of the art, our method improves the performance of elbow and wrist by 2.7% and 1.7% respectively.

Table 2: Quantitative results on FLIC dataset (PCK@0.2)

| Experiment | Elbow | Wrist |
|---|---|---|
| Toshev *et al.* [23] | 92.3 | 82.0 |
| Tompson *et al.* [21] | 93.1 | 89.0 |
| Chen and Yuille [2] | 95.3 | 92.4 |
| Ours | **98.0** | **94.1** |

## 4.3 Diagnostic Experiments

In this subsection, we conduct experiments to compare different message passing schemes, structures, and noniliear functions. The experimental results in Table 3 use the same VGG for feature extraction.

**Flooding** is a message passing schedule, in which all vertices pass the messages to their neighboring vertices simultaneously and locally as follows:

$$Q_{t+1}(\mathbf{h}_i) = \tau \left( \phi(\mathbf{h}_i) + \sum_{i' \in \mathcal{V}_{N(i)} \setminus i} Q_t(\mathbf{h}_{i'}) \otimes \mathbf{w}^{i' \to i} \right), \tag{15}$$

where $\mathcal{V}_{N(i)}$ denotes the neighboring vertices of the $i$th vertex in the graphical model. We adopt the iterative updating scheme in the work of Zheng *et al.* [29].

In Table 3, *Flooding-1itr-tree* denotes the result of using flooding to perform message passing once using CNN as in [29]. The tree structure in Fig. 2 (b) is adopted. *Flooding-2iter-tree* indicates

Table 3: Diagnostic Experiments (PCP)

| Experiment | Torso | Head | U.arms | L.arms | U.legs | L.legs | **Mean** |
|---|---|---|---|---|---|---|---|
| Flooding-1iter-tree | 93.0 | 87.5 | 73.0 | 58.9 | 84.3 | 76.4 | 76.6 |
| Flooding-2iter-tree | 93.5 | 86.7 | 73.0 | 59.8 | 83.7 | 79.0 | 77.1 |
| Flooding-2iter-loopy | 94.0 | 88.2 | 74.4 | 62.1 | 84.3 | 80.0 | 78.4 |
| Serial-tree(ReLU) | 95.5 | 88.9 | 75.9 | 63.8 | 87.1 | 81.4 | 80.1 |
| Serial-tree(Softmax) | **96.0** | **91.3** | **80.0** | **67.1** | **89.5** | **85.0** | **83.1** |

the result of using flooding to pass messages twice. The weights across the two message passing iterations are shared. Experimental results show slight improvement of passing twice than once. The result for the loopy structured graph in Fig. 2 (e) is denoted by *Flooding-2iter-loopy*. The connection of a pair of joints is decided by the following protocol: if 90% of the training sample's distance is within 48 pixels, which is the receptive field size of our filters, we connect these two joints. Improvement of 1.3% is introduced by these extra connections.

These approaches share the same drawbacks: lack of information for making predictions. With one iteration of message passing, each body part could only receive information from neighborhood parts, while with two iterations a part can only receive information from parts of depth 2. However, the largest depth in our graph is 10. Flooding is inefficient for a node to receive the messages from the other nodes. This problem is solved with the serial scheme.

**Serial** scheme passes messages following a predefined order and update information sequentially. For a tree structured graph with $N$ vertices, each vertex can be marginalized by passing the messages within $2N$ operations using the efficient sum-product algorithm [13]. The result of using serial message passing is denoted by *Serial-tree(Softmax)* in Table 3. In can be shown that the serial scheme performs better than the flooding scheme.

It is well known that softmax leads to vanishing of gradients which make the network training inefficient. In experiment, we replace $\frac{1}{z}e^{\{x\}}$ with $\beta\frac{1}{z}e^{\{\alpha x\}}$ to accelerate the training process. We set $\alpha \leftarrow 0.5$ and $\beta \leftarrow N_c$, where $N_c$ is the number of feature channels. With this slight change, the network can converge much faster than softmax without using $\alpha$ and $\beta$. The performance of using this softmax, which is derived from our CRF in (13), is 3% higher than *Serial-tree(ReLU)*, which uses ReLU as the non-linear function for passing messages among features, a scheme used in [3].

## 5   Conclusion

We propose to use CRF for modeling structured features and structured human body part configurations. This CRF is implemented by an end-to-end learning CNN. The efficient sum-product algorithm in the probabilistic model guides us in using an efficient message passing approach so that each vertex receives messages from other nodes in a more efficient way. And the use of CRF also helps us to choose non-linear functions and to know what are the assumptions and approximations made in order to use CNN to implement such CRF. The gain in performance on two benchmark human pose estimation datasets proves the effectiveness of this attempt, which shows a new direction for the structure design of deep neural networks.

**Acknowledgment**: This work is supported by SenseTime Group Limited, Research Grants Council of Hong Kong (Project Number CUHK14206114, CUHK14205615, CUHK14207814, CUHK14203015, and CUHK417011) and National Natural Science Foundation of China (Number 61371192 and 61301269). W. Ouyang and X. Wang are the corresponding authors.

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
