[Reviews · NeurIPS 2016]

Reviewer 1

Summary

A CRF-CNN framework for adding structure on top of CNNs is proposed and applied to human pose estimation. A message passing scheme for body joints, implemented with convolution between feature maps in same layer, is proposed and a neural network implementation of end-to-end learning CRF-CNN is provided.

Qualitative Assessment

1. I would like to see numbers on the MPII human pose dataset (http://human-pose.mpi-inf.mpg.de/). The LSP and FLIC datasets are older datasets with an order of magnitude less images than the MPI dataset. MPII dataset is currently the most exhaustive dataset for human pose estimation and to be called the state-of-the-art, an algorithm must outperform other algorithms on this dataset. 2. Can you show the full PCK curve instead of only the table 2? Otherwise I suspect that since your results in the high accuracy region (5-10 px) is not good, you do not show the curve. Other minor points: 1. Is K same as N? (in Algorithm 1, line 2) 2. Perhaps a better notation for softmax (\tau) can be used. Else it will seem like scalar multiple \tau. 3. The notations in the illustration (equations 16, 17, 18) seem to be confusing. It is as though the softmax operating on a scalar and not a vector. Same for the messages from the factor nodes (look like scalars). 4. The qualitative results in the supplementary are good and would like to see some of the results in the main paper as well.

Confidence in this Review

3-Expert (read the paper in detail, know the area, quite certain of my opinion)


Reviewer 2

Summary

This paper explores a CRF-CNN architecture for human pose estimation. The paper presents a an exposition of how to construct a CRF-CNN model and pass messages. Strong experimental results are provided.

Qualitative Assessment

The general idea of this work is clearly in a direction of interest to the community and the results look strong. However, there are a few aspects to this work that I find quite unclear. If they were clearer this work would have much more potential for impact. In particular, it is not clear enough if this work uses a truly 'end-to-end' approach (as stated for one of the contributions). For example on line 46 it is stated that: "We show step by step how approximations are made to use an end-to-end learning CNN for implementing such CRF model." On line 114 it is stated that "Therefore, the mapping from f to h ̃ can be implemented with one-layer transformation in a neural network and sigmoid is the activation function. h ̃ is a new feature vector derived from f and f can be obtained from lower layers in a network." However, there is no discussion about the procedure or the possibility of back propagating information into the CNN. It is stated on line 151 that "We use the fully convolutional implementation of VGG and the output of fc6 in VGG is used as the feature map". It is therefore unclear precisely what is being done with respect to this issue. * Is the CNN component being trained here through backpropagation, or is this a CRF on top of CNN features? There has been a fair amount of work that has looked at these types of random field models for human pose prior to the explosion of interest in CNNs. See the CRF work of: Kiefel, M. and Gehler, P.V., 2014, September. Human pose estimation with fields of parts. In European Conference on Computer Vision (pp. 331-346). Springer International Publishing. and the references therein for pointers to some of the other prior work. I agree that while there are a number of examples of prior work that are very close to this formulation, there is a need for work that combines CNNs and CRFs in clear and clean ways. The work presented here looks a lot like it boils down to taking features of a CNN as input into fairly well known CRF models, and therefore why not just use established inference methods like sum/max product for trees and mean field for loopy graphs ? It is also not clearly and explicitly stated why an inference procedure consisting of 'standard' sum/max product for a trees or mean-field for loopy graphs is or is not being used or compared with here. The motivation and justification of the proposed messages is unclear. These messages should be much more explicitly compared and contrasted with the corresponding established message passing equations. Ideally this comparison would be made both conceptually and empirically. The abstract asserts: "A message passing scheme is proposed, so that in various layers each body joint receives messages from all the others in an efficient way". While the manuscript is fairly clear and detailed regarding the CRF inference equations, these equations are not clearly motivated and related to the well known inference procedures of loopy BP and mean field. On line 109 it is stated that "p(h|I , Θ) in (11) is approximated by a product of independent Q(hi |I , Θ) in (13) and (14), similar to mean-field approximation." Why not use mean field exactly? In the case of loopy graphs, does the proposed algorithm become loopy BP? Since some of the motivation presented here is related to approximating h~ with an expectation (ex. (15)), is there a connection between the proposed procedure and expectation propagation? The results regarding 'flooding' vs serial updates in Table 3 seem like they should be somewhat unsurprising if viewed through the lens of formally derived mean field updates. i.e. According to mathematics of mean field, updates should be done serially - each local Q is updated based on the current Q of neighbours. If viewed through the lens of loopy BP, loopy BP has been well studied and a more detailed placement of the procedure here with known results from loopy inference should be provided. I think the paper would benefit greatly from comparing and contrasting both theoretically and empirically the proposed scheme with established mean field and loopy BP update methods. The general idea of merging CRFs and CNNs is certainly of current interest and the quality of the exposition and exploration here will have an important influence on the long term potential for this paper to have impact. The author response on these issues could alter my numerical ratings above.

Confidence in this Review

3-Expert (read the paper in detail, know the area, quite certain of my opinion)


Reviewer 3

Summary

The authors present a CRF-CNN model that is part VGG network and part CRF for human joint location estimation.

Qualitative Assessment

* Eq 2 reminds the Hidden Conditional Random Field or the Latent Dynamic Conditional random field models. In those models the partition function is calculated by belief propagation, why do you need to approximate it here? * The factorization in Eq(3) (5) (7) is not true for CRFs. If it is in your model, what are these marginals? Could you define them? * In (6), (8), (9), you use i < j for the summations but in your figures the states look adjacent, which one is it? * In many equations, eg (8) you say that \psi_{zh} is a pairwise term. Isn't it a unary term since it involves only one latent variable? * In Sect. 3 you talk about doing something similar to mean-field approximation, but that is not explained at all? Are you doing approximate inference here? It seems so from 10-14, but not explained well. * In Sect. 3 you mention that h_k are binary variables, but in Sect.2.3 you mention that h_k are categorical. Then again, in Sect. 2.1 you say that Fig. 1(a) is like a multi-layer NN, but that wouldn't be possible if h_k is not binary. Also if it's binary that's pretty similar to the Hidden-Unit Conditional Random Field, I think. * Overall, the paper might have promise, but it is poorly explained and justified, and some of the key pieces were wrong

Confidence in this Review

3-Expert (read the paper in detail, know the area, quite certain of my opinion)


Reviewer 4

Summary

This paper seeks to impose a graphical model over the latent variables within a CNN, rather than only over the output variables. The nodes in the layer before the output are arranged into groups where each group corresponds to an output variable. Each "variable" in this layer is a discrete distribution over a label space of size L. The latent variables are connected in the same manner as the output variables. Inference is performed using sum-product message passing and the marginals are provided as feature maps to the next layer.

Qualitative Assessment

Section 2 was beautifully written, and Figure 1 is very clear. However, I had some difficulty with Section 3. In eq 14, what is the input to psi_h? I previously thought that it was a labelling h_j, however now it takes Q(h_j) which is the joint probability of a labelling h_j? However, I feel that I am simply unfamiliar with the notation used in this area. I'll defer to the other reviewers on this. Is it necessary to spend so much space describing belief propagation? I didn't understand the notation with the softmax in eqs 16, 17, 18. The softmax is over the label space, but the input to the softmax function seems to be a scalar? Should it be an element-wise product instead w .* h of an inner product w' * h? Using the expectation of h as a feature map is a cool trick. I didn't understand the purpose of the approximation in eq 16. The expression for phi_h before eq 15 is confusing. Why multiply the score by the label? The next sentence discusses binary variables, in which case it makes more sense. Is the variable i in the summation of eq 14 distinct from the i in Q(h_i) or the same? It's unclear. If it is the same, then does the i < j impose an order on the groups that has an effect on the results? Perhaps you should cite the paper "Deeply Learning the Messages in Message Passing Inference" from the previous NIPS. How big are the groups (bold h_i) of latent variables (non-bold h_k) in the experiments? Is this an important parameter? I would like to know how the running times of the flooding vs serial variants compare? Does the performance of the flooding approach improve as you increase the number of iterations? Overall it felt like a good paper, however I think the clarity of Section 3 should be improved.

Confidence in this Review

1-Less confident (might not have understood significant parts)


Reviewer 5

Summary

The paper proposed a CRF-CNN based solution for human pose estimation. Experimental results using public datasets, a comparison with similar approaches in deep learning show the strength of the proposed method.

Qualitative Assessment

Overall, the paper presentation is clear except Section 3, in particular the Eq. 14-18 that are rather confusing to me. Authors seem to mix up vector and scalar together? It will be better if authors could provide the full results (currently only PCK@0.2) for the FLIC dataset since it is only based on 2 body parts.

Confidence in this Review

2-Confident (read it all; understood it all reasonably well)


Reviewer 6

Summary

This paper proposes the CRF-CNN model, it is able to take all feature-output, output-output, feature-feature relations into consideration, which is an interesting formulation for pose estimation.

Qualitative Assessment

This paper proposes the CRF-CNN model to collaboratively consider the feature-output, output-output, feature-feature relations. Which is a minor improvement over the previous researches that use at most two of such relations. Except the above mentioned flaw, the equation (14) is also quite confusing, to my understanding, bold `h_i` is a vector, Q is a scalar. So why a scalar is put into the vector position in the \psi_h term? And also Q is a positive number, while h is the hidden variable can take a real value, so this equation does not seem to be quite meaningful, or I made a misunderstanding. Further, the performance of [3] in Table 1 is not cited correctly, the authors might need a re-check. And for Table 2, [3] also provided with the performance on the FLIC dataset, but it is not cited by the authors.

Confidence in this Review

2-Confident (read it all; understood it all reasonably well)